# Formation of Pentagonal Dimples in Icosahedral Diamond Crystals Grown by Hot Filament Chemical Vapor Deposition: Approach by Non-Classical Crystallization

**Chang Weon Song [1], Rongguang Jin [2], Jung-Woo Yang [3], Nong-Moon Hwang [3,\* and Kwang Ho Kim [1,4,\***

[1] School of Materials Science and Engineering, Pusan National University, Busan 46241, Korea; cwsong@pusan.ac.kr
[2] School of Convergence Science, Pusan National University, Busan 46241, Korea; jrguang@pusan.ac.kr
[3] Department of Materials Science and Engineering, College of Engineering, Seoul National University, Seoul 08826, Korea; jwoo5432@snu.ac.kr
[4] Global Frontier R&D Center for Hybrid Interface Materials, Pusan National University, Busan 46241, Korea
\* Correspondence: nmhwang@snu.ac.kr (N.-M.H.); kwhokim@pusan.ac.kr (K.H.K.)

**Abstract:** In this study, acetone was used as a carbon source to deposit diamond films using tantalum filaments by hot filament chemical vapor deposition (HFCVD). For acetone fluxes of 80, 90, 130 and 170 standard cubic centimeters per min (sccm) and the respective hydrogen fluxes of 420, 410, 370, and 330 sccm, film thickness appeared to increase with increasing acetone, and high quality diamonds were deposited with well-defined facets of (111) and (100). For acetone fluxes of 210 and 250 sccm and the respective hydrogen fluxes of 290 and 250 sccm, however, the diamond quality was degraded with cauliflower-shaped structures evolving and the film thickness decreased with increasing acetone. The degradation of diamond quality was confirmed by Raman spectra and X-ray diffraction (XRD). Many diamond crystals grown at acetone fluxes of 80, 90, 130 and 170 sccm consisted of five (111) facets, indicating an icosahedral structure. At the corner where the five (111) facets met, there were pentagonal dimples, which implied that diamond crystals must have been etched. The decrease in film thickness at high acetone fluxes of 210 and 250 sccm also implied that the deposited film must have been etched. These results indicate that the two irreversible processes of deposition and etching occur simultaneously, which would violate the second law of thermodynamics from the classical concept of crystal growth by an individual atom. These puzzling results could be successfully explained by non-classical crystallization, where the building blocks for diamond films are nanoparticles formed in the gas phase.

**Keywords:** diamond; non-classical crystallization; icosahedral diamond; HFCVD; etching

## 1. Introduction

Since it was reported that diamonds could be synthesized at low pressure [1–3], extensive research on diamond deposition has been made. Diamond coatings have been commercialized with steady growth of their market. Although there are many methods to deposit diamonds, the most widely used one would be hot filament chemical vapor deposition (HFCVD) [4–7]. Diamond coatings by HFCVD have many advantages such as low cost and mass production.

In spite of extensive diamond CVD studies and its commercialization, there are many puzzling phenomena in the microstructure evolution or in the deposition behavior that cannot be answered easily. This is due to the lack of understanding of the deposition mechanism. Currently, it would be

beneficial to examine more rigorously some of the more puzzling morphology evolution and deposition behavior, which have either gone unnoticed or been paid little attention.

On the other hand, there has been a big issue in the crystal growth community as to the newly found growth mechanism of non-classical crystallization, where the building blocks of crystal growth are not individual atoms nor molecules, but nanoparticles [8–11]. This means that many crystals, which were believed to grow by an individual atom or molecule actually grow by nanoparticles. Non-classical crystallization has now become so established that several review papers and books [8–13] have been published. Moreover, tutorial and technical sessions devoted to this topic were included, respectively, in the spring meetings of the Materials Research Society (MRS) and the European Materials Research Society (EMRS) in 2014.

Hwang et al. [14–17] extensively studied non-classical crystallization in the chemical vapour deposition (CVD) process. They suggested that charged nanoparticles are generated in the gas phase almost without exception under typical deposition conditions of CVD. These nanoparticles then become the building blocks of thin films and nanostructures [14–18]. Non-classical crystallization in the CVD process was first suggested to explain the paradoxical experimental observation of simultaneous deposition of less stable diamond and etching of stable graphite [19]. This phenomenon violates the second law of thermodynamics if approached by the classical concept of crystal growth by an atomic unit.

Here, we studied the effect of acetone flux on the morphological evolution and the deposition behavior of diamond films. We observed some unusual features such as pentagonal dimples of the icosahedral diamond crystals and a decrease in film thickness at high acetone fluxes. These features could not be explained by the conventional concept of crystal growth by an individual atom. However, they could be explained successfully by non-classical crystallization.

## 2. Experimental Details

Acetone ($CH_3COCH_3$) was used as a carbon source, which is much more cost effective than $CH_4$ gas. Since acetone contains oxygen, tantalum filaments—which are more oxygen resistant than tungsten filaments—were used. Furthermore, the temperature of tantalum filaments could be maintained as high as 2700 K for tens of hours during HFCVD.

The schematic of the experimental HFCVD reactor is shown in Figure 1a. The temperature of the filament was calculated by computer analysis through computation fluid dynamics (CFD) code FLUNT (17.0) in ANSYS software (17.0) [20,21]. During diamond deposition, the filament temperature was measured using a 2-color pyrometer (E1RH-F1-L-0-0, Fluke, Berlin, Germany). Filament temperatures were in the range of 2500–2700 K during the HFCVD process for diamond deposition. The substrate temperature was measured by an infrared thermometer (RAYR312ML3U, Raytek, Everett, WA, USA), which fluctuates in the range of 1149–1164 K. The computational analysis was made to determine the susceptor temperature. The temperature distributions of the horizontal plane of the substrate and the vertical plane (away from the filaments) were calculated using the CFD code via ANSYS-FLUENT [20,21], and are shown respectively in Figure 1b,c. The temperature of the susceptor was estimated to be 1160 K during diamond deposition, which was within the range of the measured temperature.

The diameter and length of each tantalum filament were 0.7 mm and 32 cm, respectively. Twelve filaments were placed in parallel with an interval of 2 cm, as shown in Figure 1b. Since the temperature near the two outermost filaments was low (as shown in Figure 1b), the substrates for diamond deposition were placed within the inner 8 lines of filaments. However, even within the inner 8 lines of filaments, Figure 1b shows that the temperature is not uniform, which can result in non-uniform deposition. In order to solve this problem and to achieve uniform deposition, the substrate was rotated on the susceptor at 1 revolution per minute (rpm).

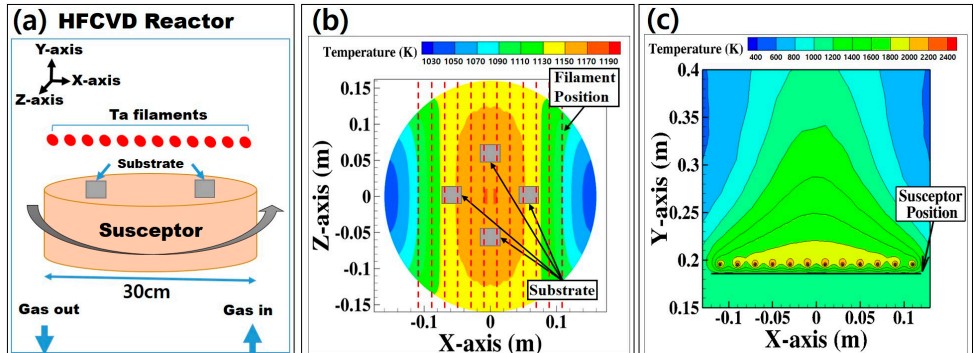

**Figure 1.** (**a**) Schematic of a hot filament chemical vapor deposition (HFCVD) reactor and temperature distributions on (**b**) the horizontal planes and (**c**) the vertical planes.

A silicon (100) wafer was used as a substrate for Field Emission Scanning Electron Microscope (FE-SEM) observation of the morphology and thickness of the diamond films with varying flux of acetone. The square silicon substrates had length and width dimensions of 3 cm × 3 cm, with a thickness of 525 ± 25 μm. Before deposition, substrates were pre-treated for diamond film formation [22–24] by diamond particles of ~500 nm being mixed with glycerin at a weight ratio of 1:1. The substrate surface was rubbed by the mixed paste to seed the diamond particles and to increase the roughness of the substrate surface.

Diamond deposition behavior was examined by varying the amount of acetone. The deposition conditions were as follows. The power applied to the filament was 16 kW, the distance between the filament and the susceptor was 1 cm and the reactor pressure was 4000 Pa, which were the optimum conditions for diamond deposition according to previous experiments [20,21]. The depositon time was 8 h. Table 1 shows the conditions for the process variables used in this experiment. To maintain the reactor pressure at 4000 Pa, the total flow rate of acetone and hydrogen was fixed at 500 standard cubic centimeters per min (sccm). For example, as the acetone flux increased from 80 to 90 sccm, the hydrogen flux decreased from 420 to 410 sccm, as shown in Table 1. Partial pressures of hydrogen and acetone in Table 1 were calculated from their flow rate using the Antoine equation.

**Table 1.** Acetone and hydrogen fluxes and their partial pressures.

| Flux of Acetone (sccm) | Flux of Hydrogen (sccm) | Partial Pressure of Acetone (Pa) | Partial Pressure of Hydrogen (Pa) |
|---|---|---|---|
| 80 | 420 | 59 | 3941 |
| 90 | 410 | 67 | 3933 |
| 130 | 370 | 97 | 3903 |
| 170 | 330 | 128 | 3872 |
| 210 | 290 | 160 | 3840 |
| 250 | 250 | 191 | 3809 |

The morphology and thickness of the deposited diamond films were analyzed by FE–SEM (S-4800, Hitachi, Tokyo, Japan). Raman spectroscopy (Jobin Yvon, Horiba, Longjumeau, France) was used to determine the diamond quality. Raman spectroscopy was operated with argon lasers at excitation wavelengths of 514.5 nm and 1800 lines/nm. XRD (D8-Advance, Bruker, Karlsruhe, Germany) was used to confirm the crystallinity of the diamond. XRD was measured under two conditions: Coupled scanning diffraction (CSD) to measure θ/2θ and sample tilting diffraction (STD) to measure 2θ with θ fixed at 3° [25]. The 2θ scan was in the range of 40° to 80°. Cu K$_\alpha$ with the wavelength of λ = 1.5406 Å was used with the sampling width of 0.02° and the scan speed of 1 deg/min. The acceleration voltage and current were 40 kV and 40 mA, respectively.

Using the Scientific Group Thermodata Europe (SGTE) Substance Database, thermodynamic calculations were performed using the Thermo-Calc software [26] for the varying amount of acetone and hydrogen used in this study. The calculations included the supersaturation ratio for precipitation of diamond and graphite, the solubility limit of diamond and graphite and the equilibrium mole fraction of diamond and graphite as a function of temperature.

## 3. Results and Discussion

Diamonds were deposited on a silicon substrate with acetone fluxes of 80, 90, 130, 170, 210 and 250 sccm, with the respective hydrogen fluxes provided in Table 1. Figure 2 shows the FE–SEM images of diamond films. For 80 sccm of acetone, the diamond does not completely cover the substrate and a continuous film is not formed, as shown in Figure 2a. This incomplete coverage of diamond on the substrate is known to occur when the substrate temperature is too low or when the distance between the susceptor and the filament is too far [20,21]. However, a continuous diamond film was formed at the same substrate temperature and the same distance between the susceptor and the filament when the flux of acetone was above 90 sccm. For example, Figure 2b–d, which depicts the cases where the acetone flux was 90, 130 and 170 sccm, respectively, shows that the diamond film was able to fully cover the substrate. Therefore, the incomplete coverage shown in Figure 2a may be attributed to the insufficient flux of acetone, which resulted in a low growth rate of diamond. More specifically, the acetone flux of 80 sccm was not enough to cover the substrate completely for the given substrate temperature and the distance between the susceptor and the filament. Both (111) and (100) facets are observed in Figure 2a–d, with their area ratio being slightly different in each figure.

In Figure 2e, we could see that the diamond grain size was noticeably reduced, although it was in the form of a diamond film. No (111) facets were found, while some (100) facets could be identified with small size diamond grains. As shown in Figure 2f, neither (111) facets nor (100) facets were observed. Instead, cauliflower-shaped structures with numerous small grains were observed.

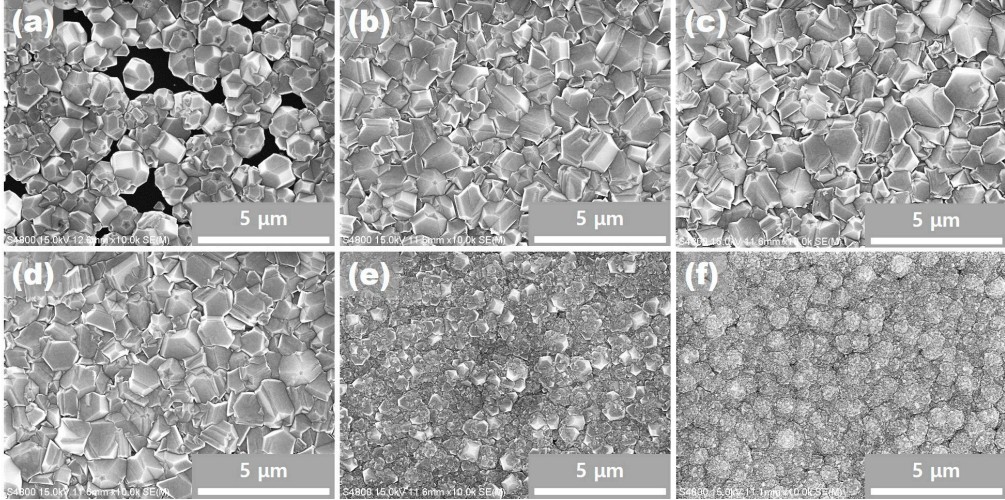

**Figure 2.** FE–SEM images of diamond films deposited at acetone fluxes of (**a**) 80, (**b**) 90, (**c**) 130, (**d**) 170, (**e**) 210, and (**f**) 250 standard cubic centimeters per min (sccm), with the respective hydrogen fluxes as shown in Table 1.

Many icosahedral structures, which consists of five (111) facets, are observed in Figure 2a–d. A noticeable feature is that most corners at which five (111) facets meet are indented and have pentagonal dimples. These pentagonal dimples are clearly revealed in the higher magnification images of Figure 3a–d. Pentagonal dimples make reentrant edges, which provide a easy site for atomic attachment. For example, a two dimensional nucleation barrier at the reentrant edges is much lower than that on the terrace. A typical reentrant edge is made by a twin at which preferential growth

occurs—it is known as a twin plane reentrant edge (TPRE) mechanism [27]. Therefore, it is difficult for pentagonal dimples to be formed by growth. Instead, it is highly probable that these dimples should have been formed by etching.

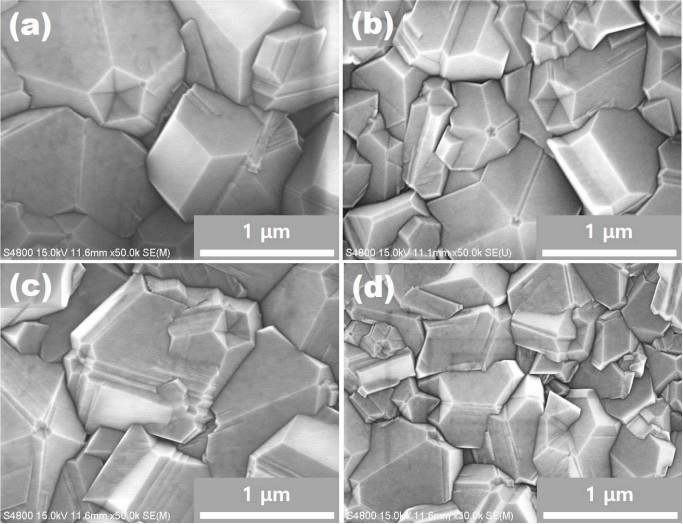

**Figure 3.** Higher magnification of Figure 2a–d, clearly showing pentagonal dimples. (**a**) 80 sccm; (**b**) 90 sccm; (**c**) 130 sccm; (**d**) 170 sccm.

Figure 4 shows how the thickness of diamond films change with varying fluxes of acetone. For acetone fluxes from 80 to 170 sccm, the thickness increased, which means that the deposition rate increased with increasing acetone flux. When the acetone flux was increased to 210 and 250 sccm, however, the thickness decreased. Normally, it is expected that the growth rate of diamond films would increase with increasing acetone flux, which is a carbon source. The decrease in film thickness for the acetone fluxes of 210 and 250 sccm implies that the films must have been etched. Therefore, Figure 4 is another piece of experimental evidence for etching.

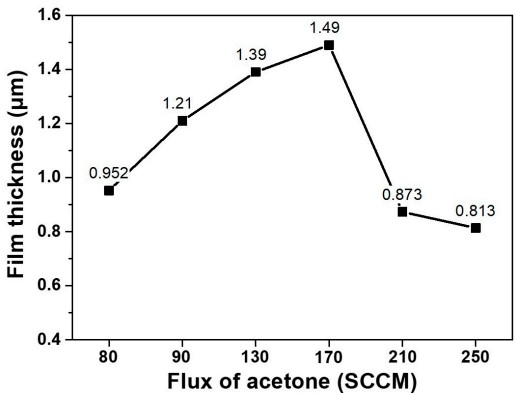

**Figure 4.** Thickness of the diamond films deposited at acetone fluxes of 80, 90, 130, 170, 210 and 250 sccm at 4000 Pa.

The etching rate must have been faster for films deposited at acetone fluxes of 210 and 250 sccm than those deposited at acetone fluxes of 80, 90, 130 and 170 sccm. This difference in the etching rate might be related with the quality of diamond films. In order to examine the quality of the deposited diamonds, the Raman spectra were measured—except in the case of the film deposited at the acetone flux of 80 sccm, as this did not fully cover the substrate. For acetone fluxes of 90, 130 and 170 sccm, the Raman spectra in Figure 5 show very clearly the characteristic peaks of diamonds at 1332 cm$^{-1}$. On the other hand, for acetone fluxes of 210 and 250 sccm, the Raman spectra show the broad D-mode

and G-mode amorphous carbon phases at 1400–1650 cm⁻¹ and a nanocrystalline diamond phase at 1090 cm⁻¹ [28,29].

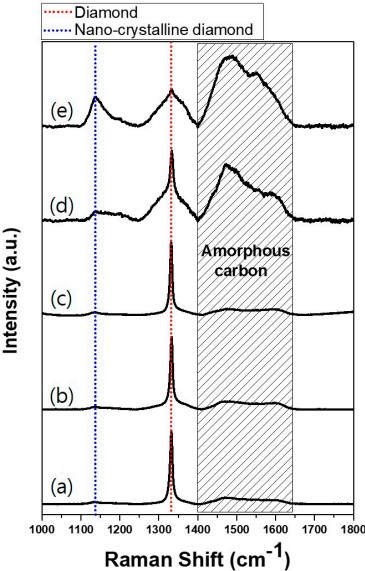

**Figure 5.** Raman spectra of the diamond films deposited at acetone fluxes of (**a**) 90, (**b**) 130, (**c**) 170, (**d**) 210, and (**e**) 250 sccm.

The phase identification of deposited films was further examined by XRD. Figure 6a shows the CSD mode XRD as a logarithmic scale. Peaks of the bare silicon wafer, which are shown in XRD pattern I, are exclusively dominant for all diamond films deposited at varying acetone flux. Since peaks of the silicon substrate are too strong compared with those of the diamond in the CSD mode, Figure 6b shows XRD in the STD mode, which removes the peaks of the silicon substrate. XRD in the STD mode of Figure 6b shows that the diamond characterization peaks are observed up to the acetone flux of 210 sccm. At the acetone flux of 250 sccm, however, a broad peak is observed at 2θ, where the diamond peak exists in other films deposited at less acetone flux, indicating that the film contains the amorphous phase with low crystallinity. This result agrees with the Raman spectra in Figure 5.

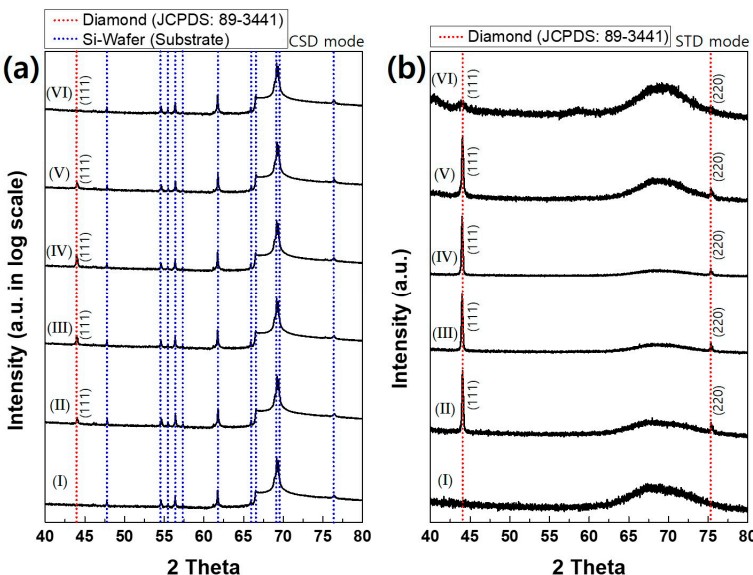

**Figure 6.** (**a**) Coupled scanning diffraction (CSD) and (**b**) sample tilting diffraction (STD) mode XRD patterns of (I) the silicon wafer and diamond films deposited at acetone fluxes (II) 90, (III) 130, (IV) 170, (V) 210 and (VI) 250 sccm.

The non-diamond or poor quality diamond deposited at acetone fluxes of 210 and 250 sccm, shown in Figures 5 and 6, would be related to the decrease in film thickness shown in Figure 4. It is highly probable that the non-diamond or poor quality diamond deposited at high acetone flux should etch much faster than the high quality diamond deposited at low acetone flux, resulting in the decrease of film thickness at acetone fluxes of 210 and 250 sccm observed in Figure 4.

In order to understand the formation of pentagonal dimples and the decrease in film thickness at high acetone fluxes of 210 and 250 sccm, it is worth making detailed thermodynamic analyses of the HFCVD process.

## 4. Thermodynamic Analysis

The thermodynamic scheme for calculating the supersaturation ratio was explained by Hwang et al. [30,31]. The supersaturation ratio for precipitation of diamond can be defined as the ratio of the partial pressure of carbon in the supersaturated gas phase to the equilibrium vapor pressure of diamond at the given temperature. Similarly, the supersaturation ratio for the precipitation of graphite can be defined as the ratio of the partial pressure of carbon to the equilibrium vapor pressure of graphite. The partial pressure of carbon in the supersaturated gas phase can be determined by Gibbs free energy minimization after excluding all condensed phases of carbon such as diamond, graphite, liquid diamond and liquid graphite. Depending on whether the supersaturation ratio is larger or smaller than unity, the driving force impels deposition or etching.

Figure 7a,b shows the supersaturation ratio for precipitation of diamond and graphite, respectively, for acetone fluxes of 80, 90, 130, 170, 210 and 250 sccm and the corresponding hydrogen flux at 4000 Pa as a function of temperature. This figure shows that the supersaturation ratio increases with increasing acetone flux. Increased supersaturation ratio means increased driving force for deposition.

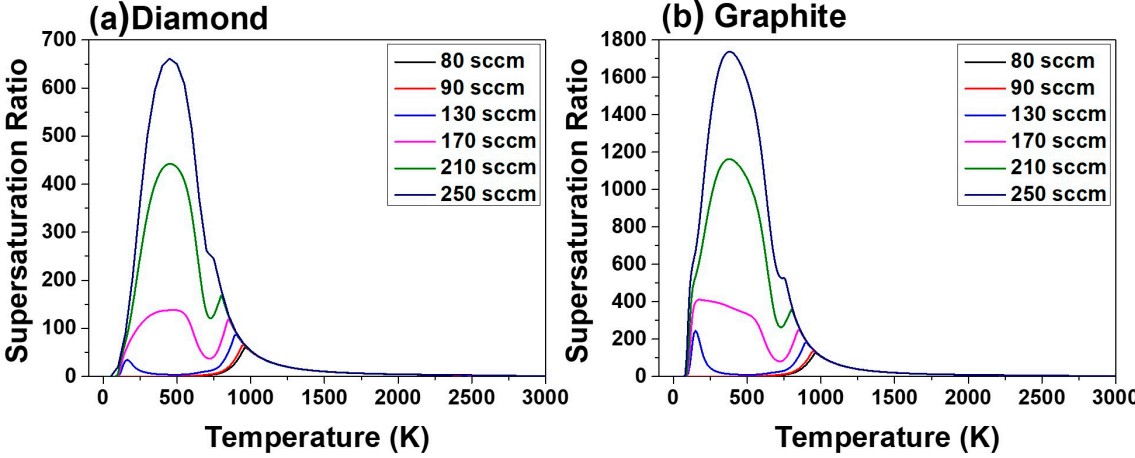

**Figure 7.** Supersaturation ratio for precipitation of (**a**) diamond and (**b**) graphite for acetone fluxes of 80, 90, 130, 170, 210 and 250 sccm at 4000 Pa as a function of temperature.

Supersaturation ratios in Figure 7a,b are larger than unity around the susceptor temperature of 1160 K, indicating that the driving force is for deposition, not for etching. Therefore, the supersaturation ratio in Figure 7 cannot explain the etching morphology of the pentagonal dimples in Figures 2a–d and 3a–d. Furthermore, the supersaturation ratio in Figure 4 cannot explain the decrease in film thickness at high acetone fluxes of 210 and 250 sccm.

From this, how can we explain the etching morphology of pentagonal dimples and the other etching-implying evidence in the form of the decrease in film thickness at high acetone fluxes of 210 and 250 sccm? It should be noted that the C–H–O system has a retrograde solubility around the susceptor temperature of 1160 K, as shown in Figure 8. This means the carbon solubility in the gas phase in equilibrium with diamond or graphite increases with decreasing temperature around 1160 K.



In this case, if diamond or graphite nucleates extensively in the gas phase, the carbon content in the gas phase will be depleted, which is determined by the solubility of carbon in the gas phase—as shown in Figure 8. If such carbon-depleted gas approaches the substrate at lower temperature, at which the solubility of carbon in the gas phase increases with decreasing temperature, the driving force would be changed to etching for both diamond and graphite. Under such a circumstance, diamond on the substrate should be etched.

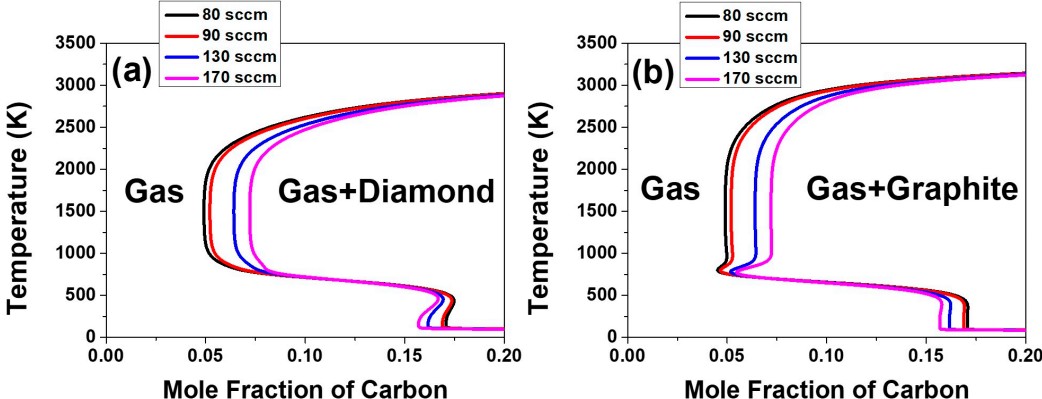

**Figure 8.** Equilibrium carbon solubility in the gas phase for (**a**) diamond and (**b**) graphite for acetone fluxes of 80, 90, 130 and 170 sccm at 4000 Pa as a function of temperature.

This aspect can be examined again by the temperature dependence on the equilibrium mole fraction of diamond and graphite in the C–H–O system in the composition range of the acetone fluxes of 80, 90, 130 and 170 sccm and the respective hydrogen flux, which is shown in Figure 9. The equilibrium mole fraction of diamond and graphite decreases slightly with decreasing temperature around 1160 K. This means that the equilibrium amount of diamond or graphite formed in the gas phase decreases with decreasing temperature. The decrease in the amount of equilibrium mole fraction would be achieved by etching.

The supersaturation ratio for precipitation of diamond in Figure 7a, as well as the equilibrium mole fraction of diamond in Figure 9a continues to increase with increasing acetone flux. Figures 7a and 9a indicate that the film growth rate should increase with increasing acetone flux. However, if the gas phase nucleation occurs and the driving force is changed to etching, the decrease in film thickness at the acetone fluxes of 210 and 250 sccm can be attributed to the increased etching rate. This is potentially because the high percentage of the non-diamond phase deposited at acetone fluxes of 210 and 250 sccm would etch much faster than the high quality diamond phase deposited at acetone fluxes of 80, 90, 130 and 170 sccm.

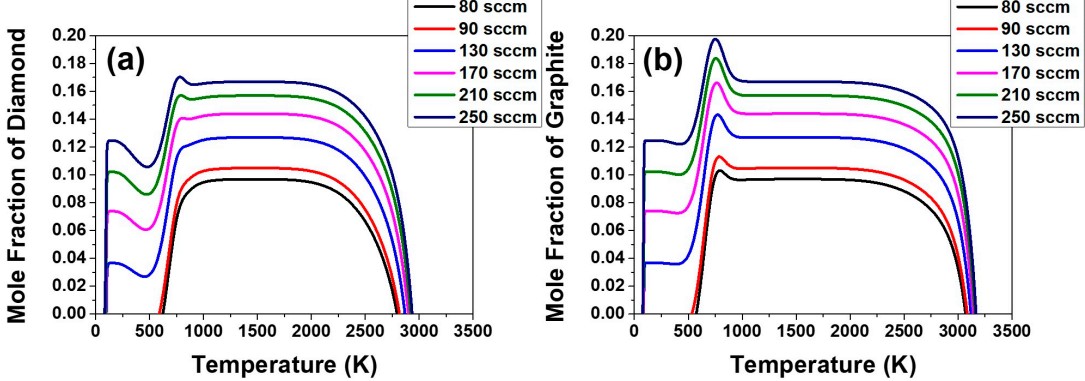

**Figure 9.** Equilibrium mole fraction of (**a**) diamond and (**b**) graphite for acetone fluxes of 80, 90, 130, 170, 210 and 250 sccm at 4000 Pa as a function of temperature.

If the pentagonal dimples are regarded as an indication of etching, the diamond crystals in Figures 2a–d and 3a–d concurrently grow and get etched. Similarly, if the decreased film thickness at high acetone fluxes of 210 and 250 sccm is regarded as an indication of etching, the films shown in Figure 2e,f grow and at the same get etched. This means that simultaneous deposition and etching of diamonds occur.

Since deposition and etching here refers to the irreversible process, the thermodynamic criterion for an irreversible transfer of carbon between the phases is the relative magnitude of chemical potential of carbon in diamond ($\mu_C^{diamond}$) and gas ($\mu_C^{gas}$). Diamond deposition means that $\mu_C^{diamond} < \mu_C^{gas}$ and diamond etching means that $\mu_C^{diamond} > \mu_C^{gas}$. Since the two inequalities cannot hold concurrently, simultaneous irreversible deposition and etching of diamond cannot occur. Otherwise, it would violate the second law of thermodynamics.

Therefore, the etching morphology of pentagonal dimples in Figures 2a–d and 3a–d, as well as the decrease in film thickness at acetone fluxes of 210 and 250 sccm in Figure 4, can be explained by the gas phase nucleation of diamond or graphite. From this, how can diamond deposit simultaneously with etching? In order to explain this phenomenon without violating the 2nd law of thermodynamics, diamond should deposit by the building block of gas phase nuclei and etch by individual atoms.

This phenomenon is closely related with a well-established observation that diamond deposition occurs simultaneously with graphite being etched [32,33]. Since, as mentioned, deposition and etching here refers to the irreversible process, it can be analyzed by the relative magnitude of chemical potential of carbon among diamond, graphite and gas. Diamond deposition means that $\mu_C^{diamond} < \mu_C^{gas}$ and graphite etching means that $\mu_C^{graphite} > \mu_C^{gas}$. Comparing the two inequalities results in $\mu_C^{diamond} < \mu_C^{graphite}$. This inequality tells that diamond is more stable than graphite, which is against the stability between diamond and graphite in the phase diagram of carbon at low pressure. Since this phenomenon of diamond deposition and simultaneous graphite etching appears to violate the second law of thermodynamics, it was often called a 'thermodynamic paradox' [34].

Hwang [19] suggested that deposition of less stable diamond and etching of stable graphite would violate the second law of thermodynamics if the deposition occurred by the atomic unit. He suggested further that in order not to violate the second law, the deposition unit of diamond should be the gas phase nuclei formed in the gas phase. If the gas phase nucleation of diamond takes place, the driving force is changed from deposition to etching according to the phase diagram of the C–H system. As a result, gas phase nucleation makes both diamond and graphite experience etching by an atomic unit. If deposition of diamond occurs by the building block of gas phase nuclei, the puzzling phenomena of simultaneous diamond deposition and etching—and of simultaneous diamond deposition and graphite etching—can be explained without violating the second law of thermodynamics.

In order for diamond growth by gas phase nuclei to be valid, diamond nuclei should be formed in the gas phase. In relation to this possibility, Hwang et al. [35–38] confirmed the gas phase generation of charged diamond nanoparticles in a series of experiments. Therefore, it can be said that the diamond films in Figures 2 and 3 grow by non-classical crystallization, where the building blocks for crystal growth are nanoparticles.

## 5. Conclusions

The morphology of pentagonal dimples as well as a decrease in film thickness at high acetone fluxes indicate that etching must have occurred while diamond films were deposited. Simultaneous deposition and etching of diamond films would violate the second law of thermodynamics unless it is assumed that diamond nucleates in the gas phase and further that the gas phase nuclei should be the building blocks of diamond growth. Therefore, the puzzling phenomena of pentagonal dimples and the decreased film thickness at high acetone fluxes could be considered evidence that diamond films in the HFCVD process grow by non-classical crystallization.

**Author Contributions:** C.W.S., K.H.K. and N.-W.H. conceived and designed the experiments; C.W.S., R.J. and J.-W.Y. performed the experiments; C.W.S. and N.-W.H. analyzed the data; K.H.K. contributed reagents/materials/analysis tools; C.W.S. wrote the paper.

**Funding:** This research received no external funding.

**Acknowledgments:** This work was supported by the Global Frontier R&D Program (2013M3A6B1078874) of the Center for Hybrid Interface Materials (HIM) funded by the Ministry of Science, ICT & Future Planning.

**Conflicts of Interest:** The authors declare no conflict of interest.

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
