# Peer review of "Formation of Pentagonal Dimples in Icosahedral Diamond Crystals Grown by Hot Filament Chemical Vapor Deposition: Approach by Non-Classical Crystallization"

_coatings, doi:10.3390/coatings9040269_

Reviewer 1 Report

The manuscript submitted by the authors is a complete study on HFCVD nanocrystal diamond growth using acetone as carbon precursor. The paper is well written and results are well supported by the experimental techniques used. I enjoyed reading the paper thanks to the clear explanation and well structured text. It could be a bit longer on the introduction, but I only missed the revolution on the rotation of the samples during the growth process. I encourage the publication of such work on Coating journal. 

Author Response

Dear Reviewer

We carefully read the comments of the referees and tried our best in revising our manuscript according to them. Our reply is in blue and the revised part is highlighted in red. All revisions are highlighted, using the "Track Changes" function in Microsoft Word.

Comments and Suggestions for Authors of reviewer 1:

The manuscript submitted by the authors is a complete study on HFCVD nanocrystal diamond growth using acetone as carbon precursor. The paper is well written and results are well supported by the experimental techniques used. I enjoyed reading the paper thanks to the clear explanation and well structured text. It could be a bit longer on the introduction, but I only missed the revolution on the rotation of the samples during the growth process. I encourage the publication of such work on Coating journal.

Our reply) In reply to this comment, we revised the section of ‘2. Experimental Details’ as follows.

Before) In order to solve this problem and to achieve uniform deposition, the substrate was rotated on the susceptor.

After) In order to solve this problem and to achieve uniform deposition, the substrate was rotated on the susceptor at 1 revolution per minute (rpm).

Reviewer 2 Report

Article is very well done, but:

Figure 2 contrast must to be improved to see better the conditions of FESEM .... scale etc.

May be is good to explain why the film deposited at the acetone flux of 80 sccm did not fully cover the substrate.

Figure 7 describe 6 curves and it can be seen only 5; why curve for 80 sccm still exists if was not complete?!

Author Response

Dear Reviewer

We carefully read the comments of the referees and tried our best in revising our manuscript according to them. Our reply is in blue and the revised part is highlighted in red. All revisions are highlighted, using the "Track Changes" function in Microsoft Word.

Comments and Suggestions for Authors of reviewer 2:

1) Figure 2 contrast must to be improved to see better the conditions of FESEM .... scale etc.

Our reply) In reply to this comment, we adjusted the contrast of Figure 2 and added a new scale bar to Figure 2.

2) May be is good to explain why the film deposited at the acetone flux of 80 sccm did not fully cover the substrate.

Our reply) Considering this comment, we added a new sentence to the section of ‘3. Results and discussion’ as follows.

Before) Therefore, the incomplete coverage in Figure 2a is attributed to the insufficient flux of acetone, which resulted in a low growth rate of diamond.

After) Therefore, the incomplete coverage in Figure 2a is attributed to the insufficient flux of acetone, which resulted in a low growth rate of diamond. More specifically, the acetone flux of 80 sccm is not enough to cover the substrate completely for the given substrate temperature and the distance between the susceptor and the filament.

3) Figure 7 describe 6 curves and it can be seen only 5; why curve for 80 sccm still exists if was not complete?!

Our reply) Thickness of lines is so thick that they are overlapped. To solve this problem, the Y axis scale and the line thickness of Figure 7 are adjusted.

Reviewer 3 Report

The comments are listed below:

1.Page 2, line 75, authors mentioned that during diamond deposition, the filament was measured by using pyrometer. Also asserted that the temperature of susceptor was difficult to measure because of rotating,  just provided the simulation results. Please clarify whether you already considered rotating influence when doing ANSYS simulation. Otherwise, please measure the substrate temperature when the susceptor is not rotating and compare it with your simulation results. 

2.Please provide more details of your susceptor and silicon substrate, such as the diameter of susceptor, size of your substrates and their positions on your susceptor, also mark their position in your figure 1 (b). 

3.Page 4, line 113, please provide the X-ray wavelength of your XRD measurement and scan rate. 

4.Page 4, Figure 2, please add clear scale bar for all your SEM images. 

5.Page 4, line 129, ‘the distance between the susceptor and the filament is too long’, here ‘too long’ should be corrected by using ‘too far’ or ‘ the distance is too large’ .

6.Page 5, Figure 3, please add clear scale bar for all your images. 

7.Page 6, please mark all the phases on your XRD peaks. 

Author Response

Dear Reviewer

We carefully read the comments of the referees and tried our best in revising our manuscript according to them. Our reply is in blue and the revised part is highlighted in red. All revisions are highlighted, using the "Track Changes" function in Microsoft Word.

Comments and Suggestions for Authors of reviewer 3:

1) Page 2, line 75, authors mentioned that during diamond deposition, the filament was measured by using pyrometer. Also asserted that the temperature of susceptor was difficult to measure because of rotating, just provided the simulation results. Please clarify whether you already considered rotating influence when doing ANSYS simulation. Otherwise, please measure the substrate temperature when the susceptor is not rotating and compare it with your simulation results.

Our reply) Considering the constructive comment, we made efforts to measure the substrate temperature and found out that the substrate temperature can be measured by an infrared thermometer (Raytek, RAYR312ML3U, USA).

That the temperature fluctuates in the range of 876 °C – 891 °C. The average temperature is 885 °C, which is 1158 K. At the rotating speed of 1 rpm, the temperature of the rotating susceptor was the same as that of the non-rotating susceptor.

We revised the section of ‘2. Experimental Details’ as follows.

Before) It is difficult to measure the susceptor temperature because it is rotated to uniformly coat the diamond. For this reason, the computational analysis is made to determine the susceptor temperature. The temperature distributions of the horizontal plane of the substrate and of the vertical plane away from the filaments were calculated using the CFD code via ANSYS-FLUENT [20, 21] and shown respectively in Figure 1b and 1c. The temperature of the susceptor was estimated to be 1160 K during diamond deposition.

After) The substrate temperature was measured by an infrared thermometer (Raytek, RAYR312ML3U, USA), which fluctuated in the range of 1149 - 1164 K. The computational analysis is also made to determine the susceptor temperature. The temperature distributions of the horizontal plane of the substrate and of the vertical plane away from the filaments were calculated using the CFD code via ANSYS-FLUENT [20, 21] and shown respectively in Figure 1b and 1c. The temperature of the susceptor was estimated to be 1160 K during diamond deposition, which is within the range of the measured temperature.

2) Please provide more details of your susceptor and silicon substrate, such as the diameter of susceptor, size of your substrates and their positions on your susceptor, also mark their position in your figure 1 (b).

Our reply) In Figure 1b, the size of the susceptor is -0.15m to 0.15m in the Z-axis. So, the diameter of the susceptor is 30 cm. Considering the comment, The substrate positions were newly added in Figure 1b and the numerical value for the susceptor size is added in Figure 1a. Besides, the information about the dimension of the silicon substrate is added in the section of ‘2. Experimental details’ as follows.

Before) A silicon (100) wafer was used as a substrate for SEM observation of morphology and thickness of diamond films with varying flux of acetone.

After) A silicon (100) wafer was used as a substrate for SEM observation of morphology and thickness of diamond films with varying flux of acetone. The square silicon substrate has the dimension of 3 cm x 3 cm with the thickness of 525 ± 25 μm.

3) Page 4, line 113, please provide the X-ray wavelength of your XRD measurement and scan rate.

Our reply) The detailed XRD conditions are added to the section of ‘2. Experimental details’ as follows.

Before) The 2θ scan was in the range of 40° ~ 80°.

After) The 2θ scan was in the range of 40° ~ 80°. Cu Kα with the wavelength of λ=1.5406 Å was used with the sampling width of 0.02° and the scan speed of 1 deg/min.

4) Page 4, Figure 2, please add clear scale bar for all your SEM images. 

Our reply) The scale bar is added.

5) Page 4, line 129, ‘the distance between the susceptor and the filament is too long’, here ‘too long’ should be corrected by using ‘too far’ or ‘ the distance is too large’.

Our reply) We changed the sentence according to the comment.

Before) The distance between the susceptor and the filament is too long.

After) The distance between the susceptor and the filament is too far.

6) Page 5, Figure 3, please add clear scale bar for all your images. 

Our reply) The scale bar is added.

7) Page 6, please mark all the phases on your XRD peaks. 

Our reply) Diamond-related planes are added. The diamond JCPDS card number is added. In the case of Si wafer, peaks matching with the bare Si wafer shown in Figure 6 I) were indicated by the dotted lines. In order to describe Figure 6a clearer, the new sentence as to the peaks of the bare silicon wafer is added to the section of ‘3. Results and discussion’ as follows.

Before) The phase identification of deposited films was further examined by XRD. Figure 6a shows the CSD mode XRD as a logarithmic scale.

After) The phase identification of deposited films was further examined by XRD. Figure 6a shows the CSD mode XRD as a logarithmic scale. Peaks of the bare silicon wafer, which are shown in the XRD pattern I), are exclusively dominant for all diamond films deposited at varying acetone flux.

Sincerely

Kwang Ho Kim
